# Biodegradation of benzo[a]pyrene by division of labor in co-culture of *Bacillus haynesii* and *Kluyveromyces marxianus* from kefir

Qing Wang,[1] Zhuonan Yang,[1] Bei Zheng,[1] Dilbar Tursun,[1] Qianjin Lv,[1] Jun Xin,[2] Rui Zhang,[1] Yanan Qin[2]

**ABSTRACT** Benzo[a]pyrene (BaP) forms during high-temperature food processing and enters the body via ingestion, raising food safety concerns. Research on degrading food's toxic substances with food-derived microorganisms is crucial for food safety, the environment, and human health. In the present study, *Bacillus haynesii* TM-41 and *Kluyveromyces marxianus* TD-3 were isolated from BaP-degrading kefir, and a co-culture system was constructed by pure culture and co-culture degradation experiments. Under the conditions of an initial BaP concentration of 19 mg/L, a temperature of 35℃, a pH value of 6.0, a degradation time of 78 h, and a final degradation rate of BaP by the co-culture system reached 71.08%. Metabolomics results showed that the TD-3 in the co-culture system degraded and utilized BaP to provide energy for its partner, TM-41. Conversely, TM-41 synthesizes and secretes specific amino acids, which are assimilated by TD-3 to fulfill its nutritional demands for growth. This study elucidated the synergistic mechanism of co-cultured microorganisms for the degradation of BaP and highlights the potential of food microbial strain resources for the removal of BaP.

**IMPORTANCE** Benzo[a]pyrene (BaP) generated during thermal food processing is of serious threat to food safety. Microbial degradation has become the preferred option for the removal of BaP due to its high efficiency, low cost, and sustainability. In our study, the degradation efficiency of benzopyrene was enhanced by constructing a co-culture system of food microbial strains. Metabolomics disclosed the bacterial-fungal synergistic degradation mechanism in this system: yeast degraded and utilized BaP to provide energy for the bacteria, while the latter supplied amino acids for the yeast to fulfill its nutritional demands for growth.

**KEYWORDS** benzo[a]pyrene, co-culture system, synergistic effect, untargeted metabolites

Benzo[a]pyrene (BaP) is a high-molecular-weight polycyclic aromatic hydrocarbon (PAH), characterized by its structure of five interconnected benzene rings, which are known for their resistance to degradation. It ranks among the top three carcinogens as identified by the World Health Organization (WHO), due to its multifaceted hazardous effects, including carcinogenic, teratogenic, and mutagenic properties (1). PAHs can enter the food supply through two primary pathways: environmental transfer to food and formation during the cooking process, where the incomplete combustion of organic matter generates these compounds. This dual origin leads to a high occurrence of PAHs in food (2). Dietary intake is the principal pathway for human exposure to PAHs (3), a factor that substantially elevates the risk of developing gastric cancer and a spectrum of cardiovascular pathologies, causing a significant health hazard (4). Consequently, the degradation of food-borne BaP is a major focus of current research.

**Peer Reviewer** Doug Cossar, Plantform Corp, Guelph, Ontario, Canada

Address correspondence to Rui Zhang, zhangrui1124@xjnu.edu.cn, or Yanan Qin, qingyalan12345@sina.com.

Qing Wang and Zhuonan Yang contributed equally to this article. The author order was determined both alphabetically and in order of increasing seniority.

The authors declare no conflict of interest.

See the funding table on p. 15.

Low-pressure cold plasma was used to degrade BaP on glass slides and in food materials. The study established that the degradation of BaP was most effective when utilizing air as the working gas (low-pressure air plasma) at a pressure of 1 Torr and 168 W. Under these conditions, the initial BaP concentration on the loaded plate was capable of being reduced by up to 82.7% within a 30-minute period (5). The application of ozone by Rozentale for the removal of BaP from smoked fish led to a 46% decrease in the concentration of BaP after a 30-minute ozonation process. However, the application of ozone also had a notable impact on the quality of the food, changing its color, flavor, aroma, and vitamin concentration (6). In comparison to physical and chemical methods, microbial degradation offers a number of advantages, including high efficiency, mild conditions, reduced cost, and the absence of secondary pollution (7). Thus, bacterial and fungal remediation has been widely used in PAHs degradation. The bacteria usually used in the removal of PAHs are *Mycobacterium, Pseudomonas,* and *Bacillus* (8), whereas fungi used are *Candida, Cryptococcus*, *Pichia,* and *Saccharomyces* (9). Probiotics are increasingly recognized for their health-promoting properties. Recent studies suggest their potential as a therapeutic strategy for the biodegradation of xenobiotics and the removal of toxic compounds (10). Sultana et al. (11) isolated five probiotic species, each with unique morphologies and the ability to tolerate BaP, from a collection of 26 fermented foods. Among these, *Bacillus velezensis* PMC10 demonstrated the highest degradation efficiency for BaP, reaching 51.32%. Bartkiene et al. (12) found that the application of lactic acid bacteria for sausage treatment before and after smoking significantly decreased both BaP and chrysene. In addition, *Bifidobacterium* sp. has been shown to possess probiotic properties, with the capacity to bind to and degrade BaP in foodstuffs, animal feed, the digestive tract, and the environment (13).

However, BaP is not the first choice of carbon source for microorganisms, as it requires high energy to utilize BaP, which limits the degradation of microorganisms to BaP. Existing strains exhibit low degradation efficiency for BaP and fail to achieve its complete mineralization (14). Harnessing the biodegradation potential of specific microbial consortia or strains effective against BaP presents a viable approach toward mitigating this issue. Compared with pure culture, co-culture has more advantages, especially the metabolic diversity, the range of degradation genes, the opportunity of complete degradation, and the special adaptability (15). Therefore, a co-culture system for degrading various pollutants has emerged as a pivotal area of research. As compared to the single-strain culture of *P. ostreatus* PO-3, co-cultures of *P. ostreatus* PO-3 with defined bacterial and nonbasidiomycete strains could enhance BaP degradation (16). Xiao et al. (17) investigated the biodegradation of BaP by *Bacillus* sp. and *Ascomycetes* sp., and the results showed that the removal efficiency of BaP by the synergistic action of fungi and bacteria reached up to 76.12% within 15 days under the mixed microbial culture. Taking into account the benefits of co-culture, the development of the high-efficiency co-culture system comprising various microorganisms represents a strategy for enhancing the degradation efficiency of BaP. This study screened and isolated a strain of *Bacillus haynesii* TM-41 and *Kluyveromyces marxianus* TD-3 from kefir. The degradation rates of the pure culture and the co-culture system were determined, respectively. The effects of various factors, such as pH, initial BaP concentration, temperature, and incubation time, and their interactions on the ability of the co-culture system to degrade BaP were investigated using response surface methodology (RSM) in conjunction with Box-Behnken. The mechanism of BaP degradation in co-culture systems of *Bacillus haynesii* TM-41 and *Kluyveromyces marxianus* TD-3 was further investigated by metabolomics.

## MATERIALS AND METHODS

### Chemicals and media

All the chemicals, BaP (>99% purity), acetone, dichloromethane, and methanol (>99% purity), were of analytical grade and purchased from Beijing Dingguo Changsheng Biotechnology Co., Ltd. The kefir grains used in this study were collected from a local household in Kashgar, Xinjiang (37°46'24"N, 75°13'27"E). Minimal salt medium (MSM; pH 7) contained $NH_4NO_3$ 1.00 g/L, $MgSO_4·7H_2O$ 0.20 g/L, $KH_2PO_4$ 0.50 g/L, $K_2HPO_4$ 1.50 g/L, NaCl 0.50 g/L, and $(NH_4)_2SO_4$ 0.50 g/L.

Since concentrations of BaP are substantially greater than its water solubility, a 1 g/L stock solution of BaP in acetone was prepared by adding BaP to a sterilized acetone solution at the ratio of 1,000 mL of acetone per gram of BaP and stored at −20°C until used.

### Kefir production

Kefir was prepared in the laboratory by inoculating low-fat (1.5%) milk with kefir grains (5 g in 100 mL milk) and incubating at 37°C for 24 h. Kefir was considered active after three successful kefir fermentation processes.

### Enrichment, isolation, and purification of microorganisms

The kefir and phosphate buffer solution were added to a centrifuge tube in the proportions of 25:25 and kept at 5,000 rpm for 20 min. The supernatant was added to the MSM containing 20 mg/L BaP, with the pH adjusted to 7.0. The culture was incubated at 37°C in a shaking incubator at 140 rpm for 72 h. The kefir (1 µL) was diluted $10^{-7}$, then inoculated on the MSM agar medium with the BaP solution, and incubated at 37°C for 3 days. Single colonies of different forms were selected, and pure cultures were obtained. The strains were stored in MRS medium containing 50% glycerol at −80°C for subsequent use.

### Initial identification of the benzopyrene-degrading microorganisms

TM-41 and TD-3 were successively identified by colony morphology, microscopic morphology, and 16S rRNA or ITS sequence analysis, respectively. DNA was extracted using the bacterial and yeast genomic DNA extraction kit (Solarbio Bio Inc., China) and amplified by PCR. The bacterial 16S rRNA region (27F and 1492R) and the fungal ITS2 region (ITS1 and ITS4) were amplified using universal primers, respectively. The purified PCR products were sent to Hangzhou Youkang Biotechnology Co., Ltd (Hangzhou, China) for sequencing. The NCBI Blast program was employed to retrieve homologous sequences, and a phylogenetic tree was constructed via MEGAX software with the neighbor-joining method.

### Biodegradation studies

#### Pure culture experiments

For preparation of cell suspension, *Bacillus haynesii* TM-41 and *Kluyveromyces marxianus* TD-3 were transferred to MRS medium for overnight culture. The bacteria and fungi were collected after centrifugation and washed with phosphate buffer solution three times, and then the cell suspension was adjusted to $OD_{600} = 1$.

The degradation capacity of the bacteria and fungi was evaluated by the shake flask method. Ten milliliters of cell suspension was added to 90 mL of the MSM containing 20 mg/L BaP to obtain a pure culture for TM-41 and TD-3. The control group was not added to the cell suspension. The cultures were incubated at 37°C in an orbital shaker at 150 rpm for 5 days. Each set of experiments was run at least in triplicate.

## Co-culture experiments

Ten milliliters of cell suspension of both strains (fungi:bacteria = 2:1) was added to 90 mL sterile MSM supplemented with BaP (20 mg/L) to obtain co-cultures. Experimental conditions were the same as for the biodegradation studies using pure strains as previously described above. All experiments were conducted in triplicate.

The degradation of BaP was measured by high-performance liquid chromatography (HPLC) with 1 mL of culture taken at different times (0, 8, 16, 24, 48, 72, 96, and 120 h) in pure culture and co-culture systems. The optical density at 600 nm of the strains was measured with a UV-visible spectrophotometer to determine cell growth, and $OD_{600}$ was used to represent biomass. An equal volume of dichloromethane was added to the culture, which was shaken for 30 min and centrifuged for 10 min (8,000 rpm). Then, the organic phase was collected, the extraction was repeated twice, and the extracts were combined, filtered by an organic phase filter membrane with a pore size of 0.22 µm, and detected using HPLC.

HPLC conditions were as follows: C18 Diamosil TM reverse phase column, chromatographic column (250 mm × 4.6 mm, particle size 5 µm); mobile phase, pure methanol/water (volume ratio 100/0); UV detector, wavelength 254 nm; injection volume, 20 µL; column temperature, 34°C; retention time, 10 minutes. The degradation rate was calculated using the following equation:

$$\text{Rd} = (C_0 - C_n)/C_0 \times 100\%$$

where Rd (%) represents the degradation rate, $C_0$ (mg/L) represents the initial concentration of BaP, and $C_n$ (mg/L) represents the remaining concentration of BaP after incubation for $n$ hours.

## Biodegradation of BaP by a co-culture system and analysis of influencing factors

The influencing factors, including initial BaP concentration, temperature, pH, and incubation time, were analyzed to determine their impact on degradation efficiency. Keeping other conditions constant, initial BaP concentrations of 10–60 mg/L, temperatures of 27°C–42°C, pH level of 3–10, and incubation times of 0–120 h. The samples were collected at various time points to determine the concentrations of BaP and cells.

## Optimization of degradation conditions by RSM

The main and interaction effects of factors, including initial BaP concentration, temperature, pH, and incubation time, on BaP degradation conditions of the co-culture system were optimized using RSM with a quadrivariate Box–Behnken design performed in statistical analysis software (Design-Expert 8.0.6). The study consisted of 29 experimental runs, with five replicates at the center point. Based on a preliminary study, four critical factors and their optimal ranges were selected in this experiment: initial BaP concentration (10–30 mg/L), temperature (27°C–37°C), pH (5–7), and incubation time (60–84 h) (Table S1).

## Metabolites extraction and UHPLC-QE-MS/MS analysis

Five milliliters of liquid culture from the pure and co-cultures of TM-41 and TD-3 were collected and stored at −80°C. Samples collected at the early (24 h), middle (48 h), and late (72 h) stages of degradation were designated as M24, D24, and MD24; M48, D48, and MD48; and M72, D72, and MD72, respectively. Five independent biological replicates were used for the metabolomics analysis. The samples were slowly thawed at 4°C. A 100 µL sample from each liquid culture was extracted with 400 µL methanol/acetonitrile/water (2:2:1, vol/vol) solutions, followed by vortexing and a 30-minute low-temperature sonication. For protein precipitation, the samples were kept at −20°C for 10 min and then centrifuged at 14,000 × $g$ at 4°C for 20 min. The supernatant was collected

and vacuum-dried. Furthermore, to prepare the samples for UHPLC-MS/MS analysis, the dried extract was reconstituted in 100 µL of acetonitrile/water (1:1, vol/vol), vortexed, and centrifuged at $14,000 \times g$ at 4°C for 15 min to obtain the clear supernatants and then transferred to sample vials for further analysis.

The intermediate products of BaP degradation were identified by UHPLC-QE-MS/MS (Thermo Fisher Scientific). A sample volume of 2 µL was subjected to separation using an ACQUITY BEH Amide column (2.1 mm× 100 mm, 1.7 µm) and subsequently analyzed using mass spectrometry (MS) detection. The mobile phases were as follows: A, water containing 25 mM ammonium acetate and 25 mM ammonium hydroxide; B, acetonitrile. The gradient elution program was as follows: 0–1.5 min, 2% A and 98% B; from 1.5 to 12 min, 2% A and 98% B to 98% A and 2% B; from 12 to 14 min, 98% A and 2% B; from 14.0 to 14.1 min, 98% A and 2% B to2% A and 98% B); from 14.1 to 17 min, 2% A and 98% B. The injection volume of the sample (2 µL), flow rate (0.5 mL/min), and column temperature (25°C) were maintained. The ion source, electrospray ionization (ESI), was operated positively or negatively. The electrospray ionization source temperature was set at 600°C, with a mass spectrometry voltage of 5,500 V (positive) and −5,500 V (negative). The gas source I (GS I) was maintained at 60 psi, gas source II (GS II) at 60 psi, and curtain gas (CUR) at 30 psi. The MS and MS/MS resolutions were 60,000 and 30,000, respectively. The detection was carried out at the mass range of 70–1,200 $m/z$.

## Data processing and statistical analysis

All statistical analyses were performed using IBM SPSS Statistics, version 19.0. Student's $t$-test was used for comparisons between two groups, and one-way analysis of variance (ANOVA) was used for comparisons among multiple groups. The standard deviation was expressed by the error bars of three repeated experiments. The metabolome data were analyzed on the free online platform of Wekemo Bioincloud (https://www.bioin-cloud.tech).

## RESULTS

### Isolation and characterization

Bacterial and fungal strains were obtained from kefir that degrades BaP and were named TM-41 and TD-3, respectively. Its morphology and phylogenetic tree analysis are shown in Fig. 1. Colonies of strain TM-41 grown on MRS agar plates were creamy-white, slightly raised, folded surfaces, and irregular edges (Fig. 1a). The strain was stained purple with Gram dye and identified as a Gram-positive species (Fig. 1b). Strain TD-3 showed a slightly protruding cream-colored colony with a smooth surface and clean edges (Fig. 1c). After staining with methylene blue, the organism was colorless and the cells were oval in shape (Fig. 1d). The phylogenetic tree analysis of strain TM-41 and TD-3 based on 16S rDNA and ITS sequences is shown in Fig. 1e and f, respectively. TM-41 had the highest similarity of 99.52% with the *Bacillus haynesii* NOK58 strain, and TD-3 had the highest similarity of 99.59% with the *Kluyveromyces marxianus* AUMC 7259 strain. Thus, they were named *Bacillus haynesii* TM-41 and *Kluyveromyces marxianus* TD-3.

### Establishment of a fungal-bacterial co-culture system for degrading BaP

The degradation of BaP by TM-41 and TD-3 pure culture and the co-cultured was assessed at a concentration of 20 mg/L (Fig. 2a). At 96 hours, the pure culture had the highest biodegradation rate for BaP, at 36.79% and 38.63%, respectively. However, the observed degradation potentials of the co-cultures ranged from 19.87% to 44.54%, and the highest level of degradation was achieved at 72 h. The co-culture significantly enhanced BaP biodegradation in comparison with the pure TM-41 or TD-3 culture. As shown in Fig. 2b, strain TM-41 and strain TD-3 reached the maximum $OD_{600}$ values of 1.04 and 1.27 at 120 h and 96 h, respectively. Compared to pure culture, the growth of co-cultured microorganisms continuously increased during the reaction period and

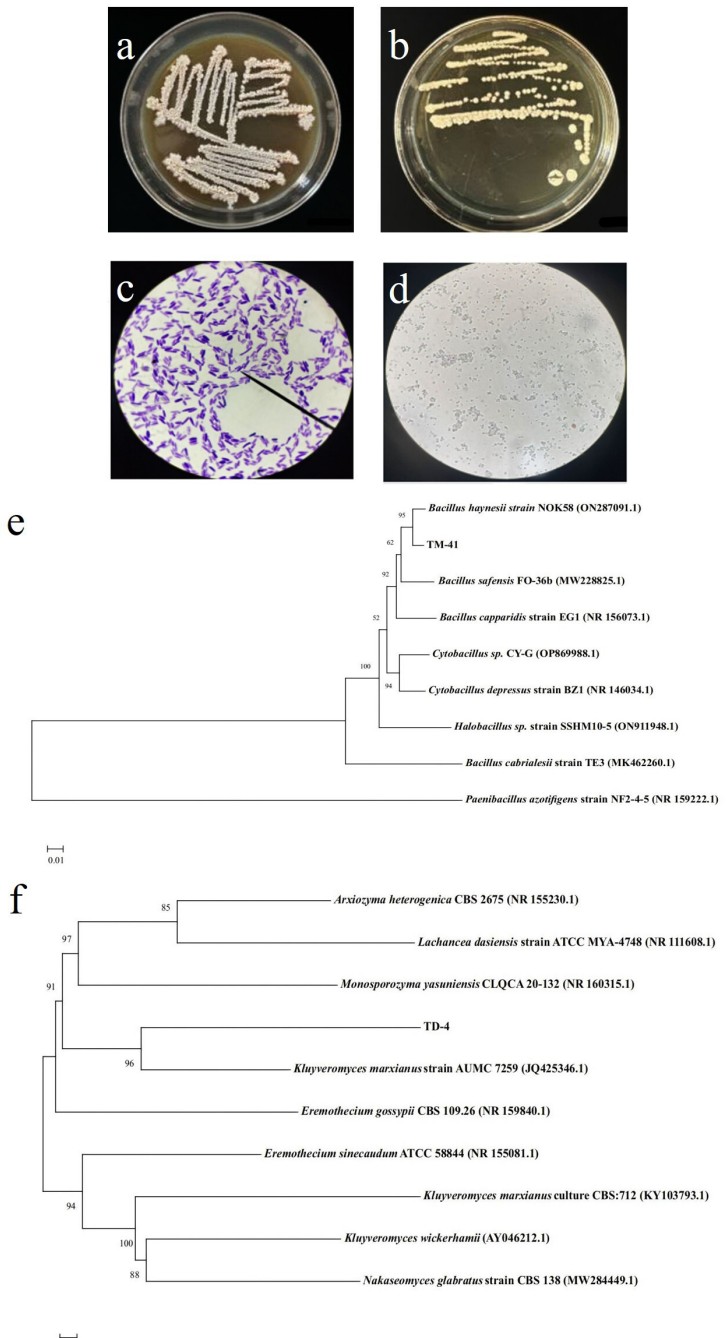

**FIG 1** Colony of the strain TM-41 (a) and TD-3 (b) on MRS medium. (c) Microscopic gram staining of strain TM-41. (d) Microscopic methylene blue staining of strain TD-3. (e) Neighbor-joining phylogenetic trees based on 16S rDNA sequences of strain TM-41. (f) The phylogenetic tree based on ITS sequences of strain TD-3.

remained higher than that of pure culture. Thus, corroborating the highest BaP biodegra-dation (44.54%) obtained for the co-culture system.

## Optimization of the co-culture system for BaP degradation

During the degradation process, many environmental factors influence the activity of bacteria and fungi in BaP degradation, including both abiotic and biotic factors (18). Hence, this study investigated the key abiotic factors influencing the co-cultured

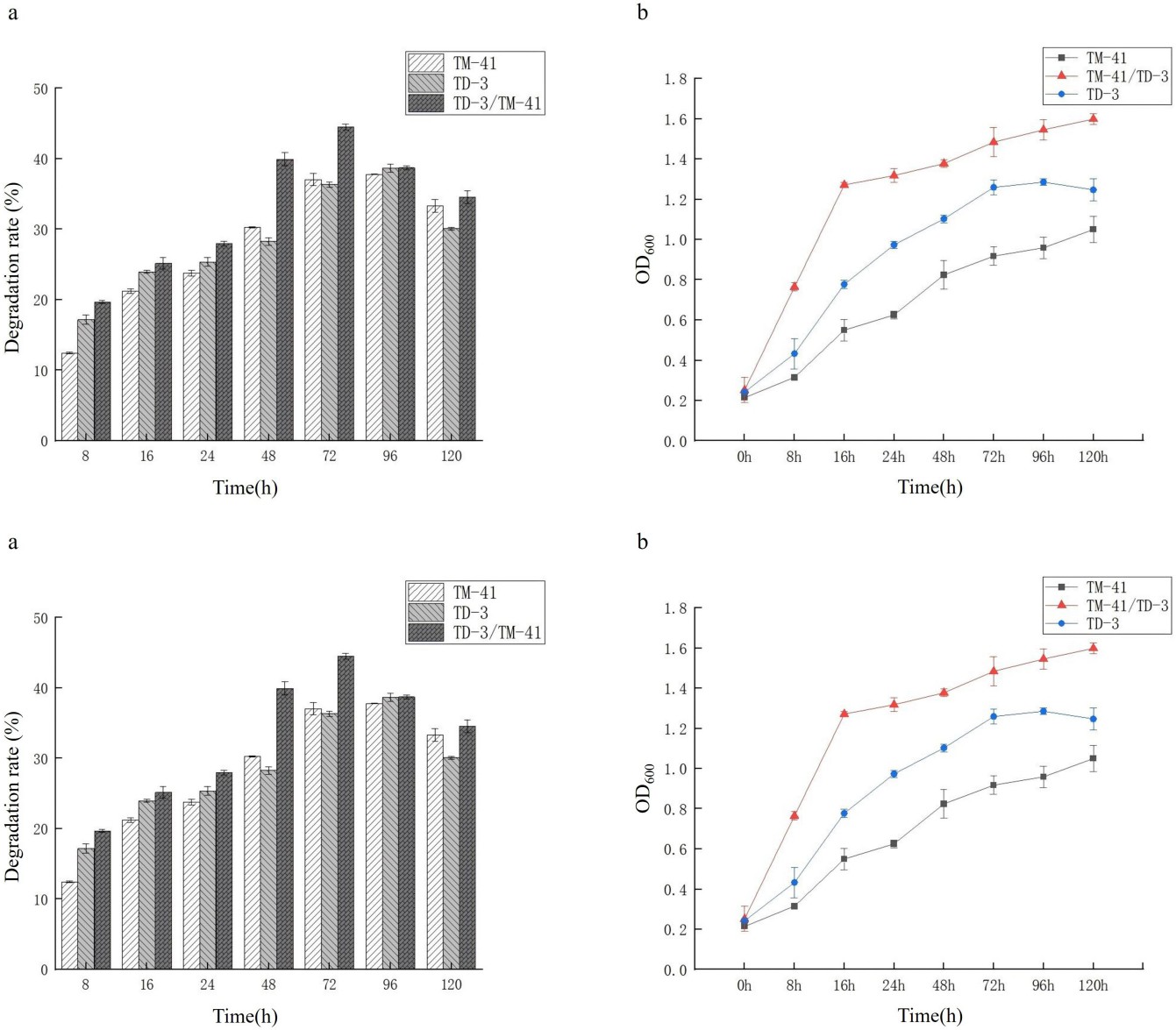

**FIG 2** (a) BaP degradation of pure culture and co-culture with time. Error bars represent the standard error of three replicate experiments. (b) OD$_{600}$ of pure culture and co-culture with time. Error bars represent the standard deviation of three replicate experiments.

microbial degradation of BaP, including initial concentration, temperature, pH, and incubation time. The concentration of BaP experiment showed that the degradation efficiency decreased with an increase in concentration, and the highest degradation efficiency was 58.10% when the concentration of BaP was 20 mg/L (Fig. 3a). Concurrently, the OD$_{600}$ also peaked at 1.8. However, the degradation rate and OD$_{600}$ of the co-culture system significantly decreased when the initial concentration of BaP was higher than 20.00 mg/L ($P < 0.05$). The results of the effect of temperature on the degradation of BaP by the co-culture system are shown in Fig. 3b. The optimal growth temperature was 37°C, with the highest value of OD$_{600}$ at 1.56, and the degradation rate was achieved at 57.22% ($P < 0.05$). The degradation efficiency decreased under acidic or alkaline conditions (Fig. 3c). The co-culture system exhibited enhanced BaP degradation when the pH value is in the range of 5–7, with a degradation rate of 67.8% at pH 6.0 and the concentration reached an OD$_{600}$ value of 1.65, which was the optimal pH for both degradation and growth ($P < 0.05$). Figure 3d illustrates that the degradation rate initially increased and subsequently decreased during the incubation time. A maximum degradation

efficiency of 63.18% was achieved at 84 hours, a value that was statistically significantly higher than observed at all other time points ($P < 0.05$). These results showed that initial BaP concentration, temperature, pH, and incubation time had significant effects on the degradation efficiency.

The Box–Behnken design was applied to determine the effects of important variables, including BaP concentration ($A$), temperature ($B$), pH ($C$), and time ($D$), according to previous single-factor experiments. The experimental design and the response of dependent variables for the degradation efficiency of BaP ($Y$) are presented in Table S2. Data from Table S2 were processed by the response surface regression procedure, and the results were obtained by fitting with the second-order polynomial model (equation 1):

$$Y = 66.54 - 1.85A + 6.79B - 1.17C + 18.94D - 0.21AB - 0.17AC - 2.98AD + 2.83BC + 2.17BD + 0.96CD - 15.17A^2 - 10.28B^2 - 9.63C^2 - 15.36D^2 \tag{1}$$

where $Y$ is the BaP degradation rate (%), $A$ is the BaP concentration (mg/L), $B$ is the temperature (°C), $C$ is the pH, and $D$ is the time (h). The ANOVA result of the regression model is exhibited in Table S1. The high determination coefficient $R^2$ of 0.9309 suggests that the model effectively captured around 93% of the responses, indicating strong agreement between the predicted and experimental values. In general, this model for BaP degradation is highly significant ($P < 0.0001$), suggesting that the quadratic polynomial model developed for BaP degradation by co-cultures was reliable and workable in representing the actual relationship between the response and variables.

Design-Expert 8.0.6 software was used to plot the response surface of the test results. The three-dimensional surface plots and contour plots of the combined effects of initial concentration of BaP and time, shown in Fig. 4a and b, respectively, indicate that the rate of BaP degradation initially increases and subsequently decreases with an increase in the initial concentration of BaP. Furthermore, the degradation rate of BaP by the co-culture system increased with time. The 3D surface and contour plots in Fig. 4c and d showed that reducing the pH from 8.0 to 6.0 at a temperature of 32°C significantly increased the degradation rate from 15.98% to 68.94%. The contour plots indicate that the two factors interact with each other. The effect of pH on the degradation of BaP by the co-culture system was greater than the effect of temperature.

The optimal conditions for the optimal analysis of the established mathematical model were an initial BaP concentration of 19.22 mg/L, a temperature of 32.93°C, a pH of 6.04, and an incubation time of 78.16 h, at which time the model predicted that the degradation rate of BaP was 73.48%. Based on the optimized conditions from the predicted results, an initial BaP concentration of 19 mg/L, a temperature of 35°C, a pH of 6.0, and an incubation time of 78 h were used as the optimal degradation conditions for the validation test. The BaP degradation rate was 71.08%, which was 26.54% higher than before optimization (44.54%), and the relative error was small compared with the predicted value, indicating that the BaP degradation conditions optimized by the response surface method were true.

## Non-targeted metabolomic profiling during BaP degradation

Metabolic differences between the treatments were assessed using partial least squares discriminant analysis (PLS-DA) and principal coordinate analysis (PCA), which provided insights into the overall variability of the respective metabolite profiles (19). The orthogonal PLS-DA (OPLS-DA) score plot exhibited a distinct separation in the metabolites among the MD24, MD48, and MD72 groups along component 1 (31.2%) and component 2 (13.3%) (Fig. 5a). Similarly, the PC score plot illustrated a clear separation across groups with PC1 (36.90%) and PC2 (9.90%), suggesting that the metabolomics profile of the co-culture system was significantly altered at distinct phases (Fig. 5b). Furthermore, Fig. 5c through e illustrates the significant upregulation and downregulation of metabolites in the differential groups, providing information on the specific

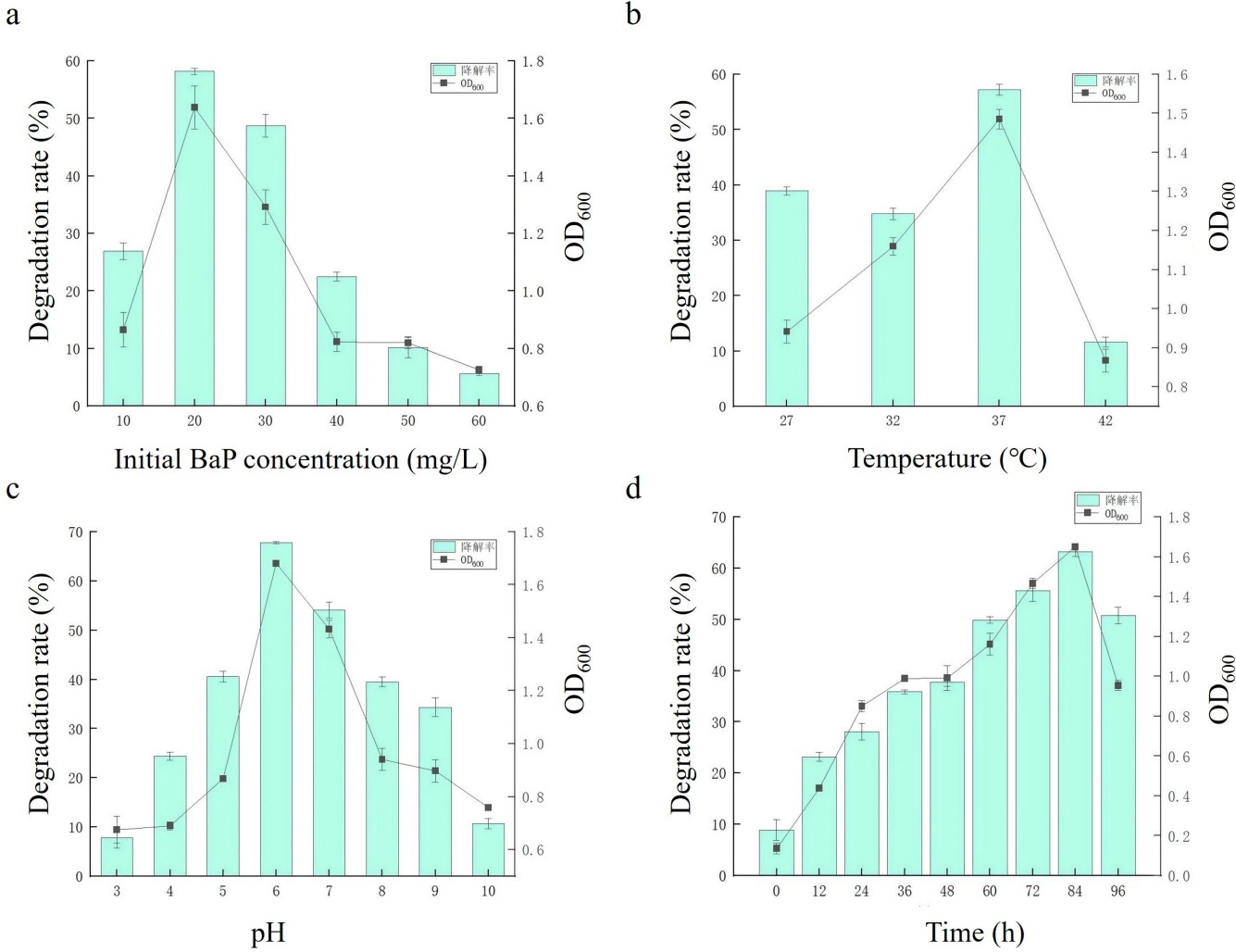

**FIG 3** Effect of initial BaP concentration (a), temperature (b), pH (c), and time (d) on degradation of BaP by the TD-3 and TM-41 co-culture system.

metabolomic changes induced by the co-cultured microorganisms at distinct time points (24, 48, and 72 h). For instance, in the MD24 vs MD48 group, 20 metabolites were identified as significant, with 12 being upregulated and 8 downregulated (Fig. 5c). Figure 5d shows the MD48 vs MD72 group displayed 133 significant metabolites, including 68 upregulated and 65 downregulated, whereas the MD24 vs MD72 group expressed 179 significant metabolites, with 79 upregulated and 100 downregulated (Fig. 5e).

Differentially expressed metabolites were identified based on two criteria: a variable importance in projection (VIP) score greater than 1.0 from the OPLS-DA model, and a statistical significance of $P < 0.05$ from the $t$-test (20). Figure 6a through c illustrates the top 20 differential metabolites, each highlighting VIP >1 and $P < 0.05$ across all treatment groups. Comparison of the MD24 and MD48 groups revealed 20 distinct metabolites. Among these, 12 metabolites exhibited an increase in the MD48 group compared with the MD24 group; 8 metabolites showed a decrease in the MD48 group. Notably, the metabolite content identified by MD48 was significantly different from that of MD24, including lipids and lipid-like molecules (50.00%), organoheterocyclic compounds (25.00%), organic acids and derivatives (20.00%), and nucleic acids (5.00%) (Fig. 6a). Moreover, most metabolites were similarly expressed in the MD24-vs-MD72 and MD48-vs-MD72. Specifically, 10 metabolites were found to be upregulated in the MD24 group compared to the MD72 group, while the same 10 metabolites were also upregulated in the MD48 group relative to MD72, including lipids and lipid-like molecules (27.07%),

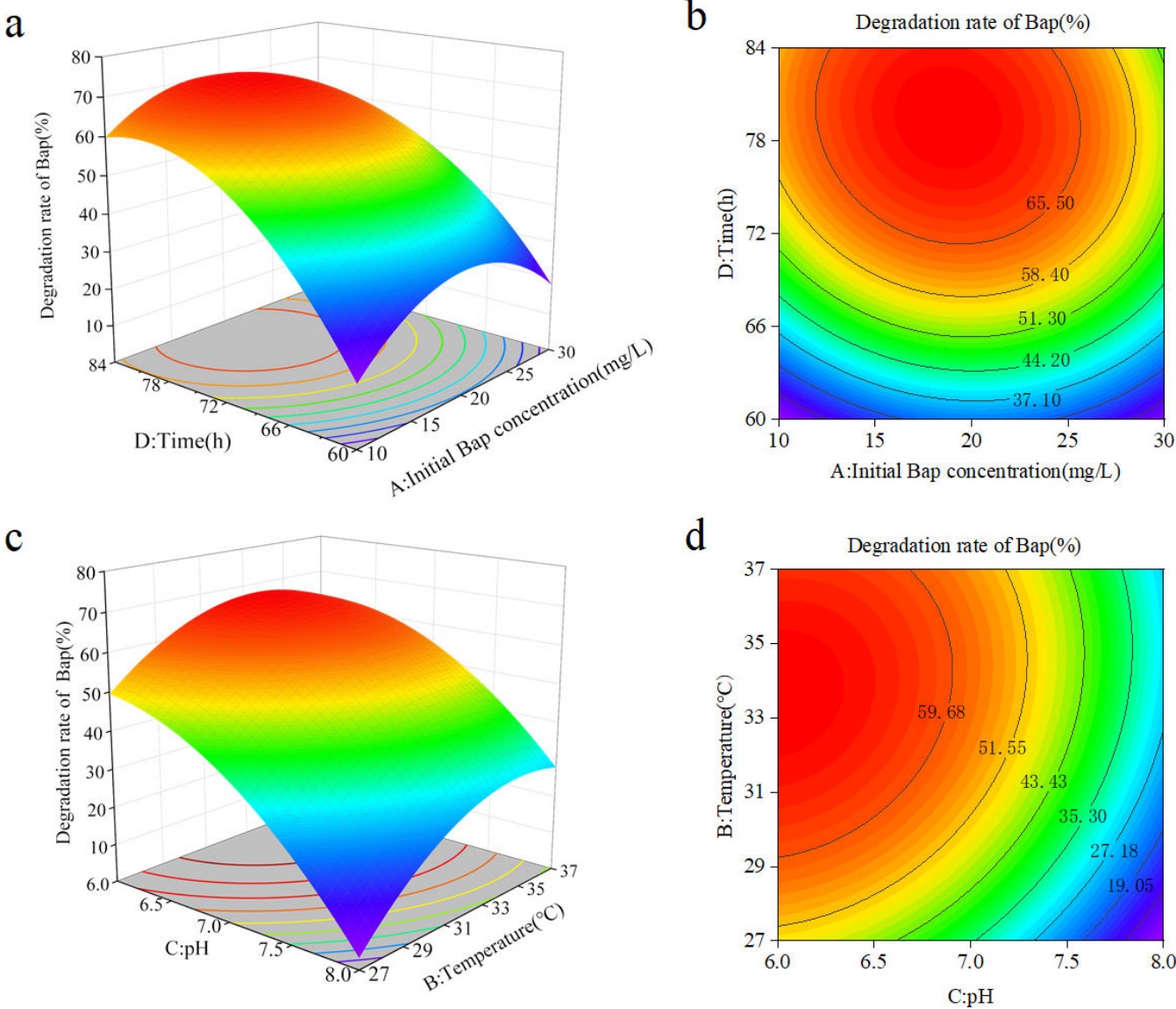

**FIG 4** 3D response surfaces (a and c) and contour plots (b and d) illustrating the degradation of BaP as influenced by time, initial BaP concentration, pH, and temperature.

organic acids and derivatives (20.30%), organoheterocyclic compounds (12.03%), benzenoids (10.53%), nucleic acids (10.53%), organic oxygen compounds (8.27%), organic nitrogen compounds (4.51%), phenylpropanoids and polyketides (2.26%), alkaloids and derivatives (1.50%), organometallic compounds (1.50%), nucleosides, nucleotides, and analogs (0.75%), and lignans, neolignans, and related compounds (0.75%) (Fig. 6b and c).

Figure S1a illustrates the metabolites with significant content differences between M72 and MD72, including organic acids and their derivatives (28.79%), organoheterocyclic compounds (18.18%), benzenoids (12.12%), nucleic acids (12.12%), organic nitrogen compounds (9.09%), lipids and lipid-like molecules (6.06%), phenylpropanoids and polyketides (6.06%), alkaloids and derivatives (4.55%), organic oxygen compounds (1.52%), and lignans, neolignans, and related compounds (1.52%). In comparison with MD72, D72 revealed metabolites with significantly altered contents, particularly in the

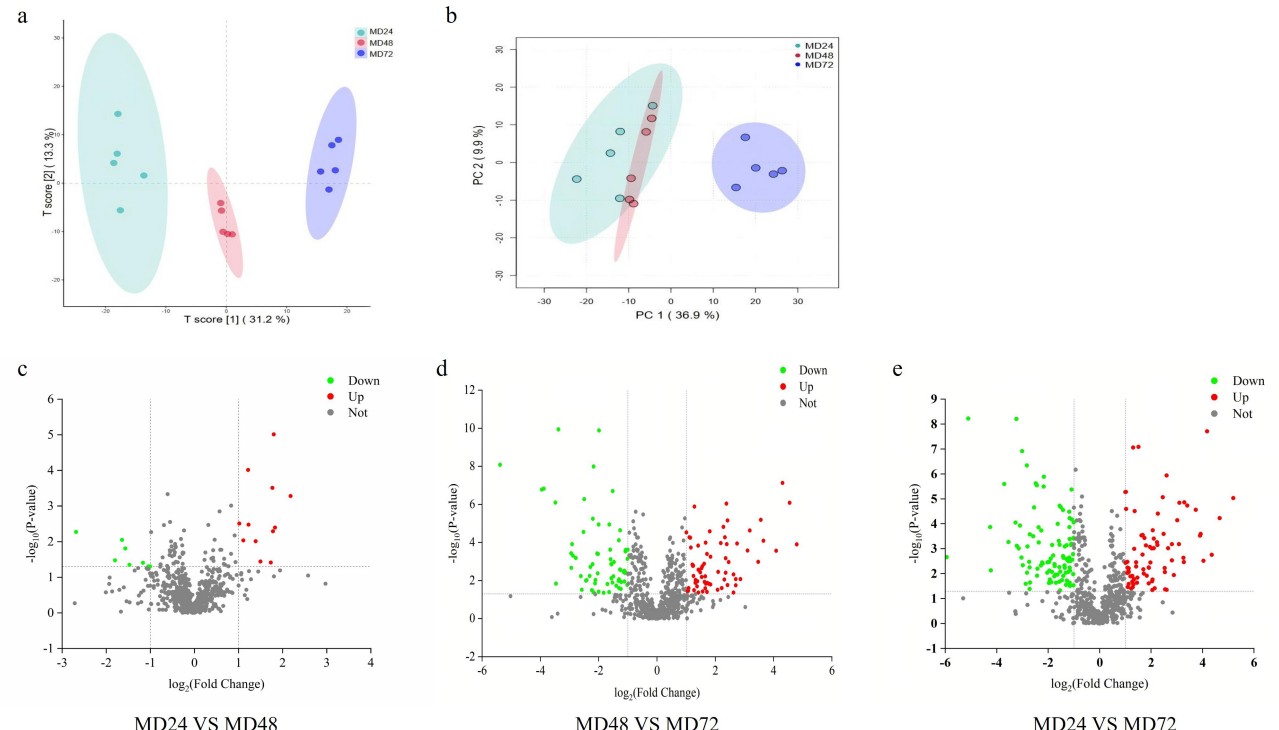

**FIG 5** (a) The OPLS-DA score plot shows a clear separation degree of the three groups of samples. (b) PCA also demonstrates the separation degree in all the groups. (c–e) The horizontal coordinate of volcano graphs represents the value of fold change of metabolite expression difference between two groups, that is, log2FC, and the vertical coordinate is the value of statistical test of metabolite expression difference, that is, −log10 ($P$ value); the higher the value, the more significant the expression difference. The values of horizontal and vertical coordinates are logarithmized. The points on the green colored are metabolites with downregulated expression differences, and the points on the red colored are metabolites with upregulated expression differences.

category of lipids and lipid-like molecules (21.43%), benzenoids (14.29%), organoheterocyclic compounds (14.29%), organic nitrogen compounds (14.29%), and nucleic acids (7.14%) (Fig. S1b).

Pathway analysis of differential metabolites elucidated the main metabolic process changes during BaP degradation by the co-culture system. Additionally, Groups MD24 vs MD48, MD48 vs MD72, and MD24 vs MD72 involved 28, 81, and 75 metabolic pathways, respectively, which indicated that the number of co-culture metabolic pathways increased with incubation time. Significant differences between groups MD24 vs MD48 were seen for aminoacyl-tRNA biosynthesis (map00970), nucleotide metabolism (map01232), as well as tryptophan metabolism (map00380) (Fig. 7a). The most abundant metabolic pathways involved in groups MD48 vs MD72 included biosynthesis of amino acid (map01230), arginine and proline metabolism (map00330), aminoacyl-tRNA biosynthesis (map00970), histidine metabolism (map00340), lysine degradation (map00310), steroid hormone biosynthesis (map00140), D-amino acid metabolism (map00470), glycine, serine, and threonine metabolism (map00260), glycerophospholipid metabolism (map00564), biosynthesis of various other secondary metabolites (map00997), sphingolipid metabolism (map00600), tryptophan metabolism (map00380), arginine biosynthesis (map00220), and beta-alanine metabolism (map00410) (Fig. 7b). Additionally, significant differences between groups M72 vs MD72 were seen for glycerophospholipid metabolism (map00564), histidine metabolism (map00340), arginine and proline metabolism (map00330), as well as beta-alanine metabolism (map00410) (Fig. 7c). Differential metabolic pathways in the D72 vs MD72 groups included glutathione metabolism (map00480), butanoate metabolism (map00650), aminoacyl-tRNA biosynthesis (map00970), arginine and proline metabolism (map00330), and steroid hormone biosynthesis (map00140) (Fig. 7d). Notably, amino acid biosynthesis exhibited as a differential metabolic pathway between bacteria and coculture system,

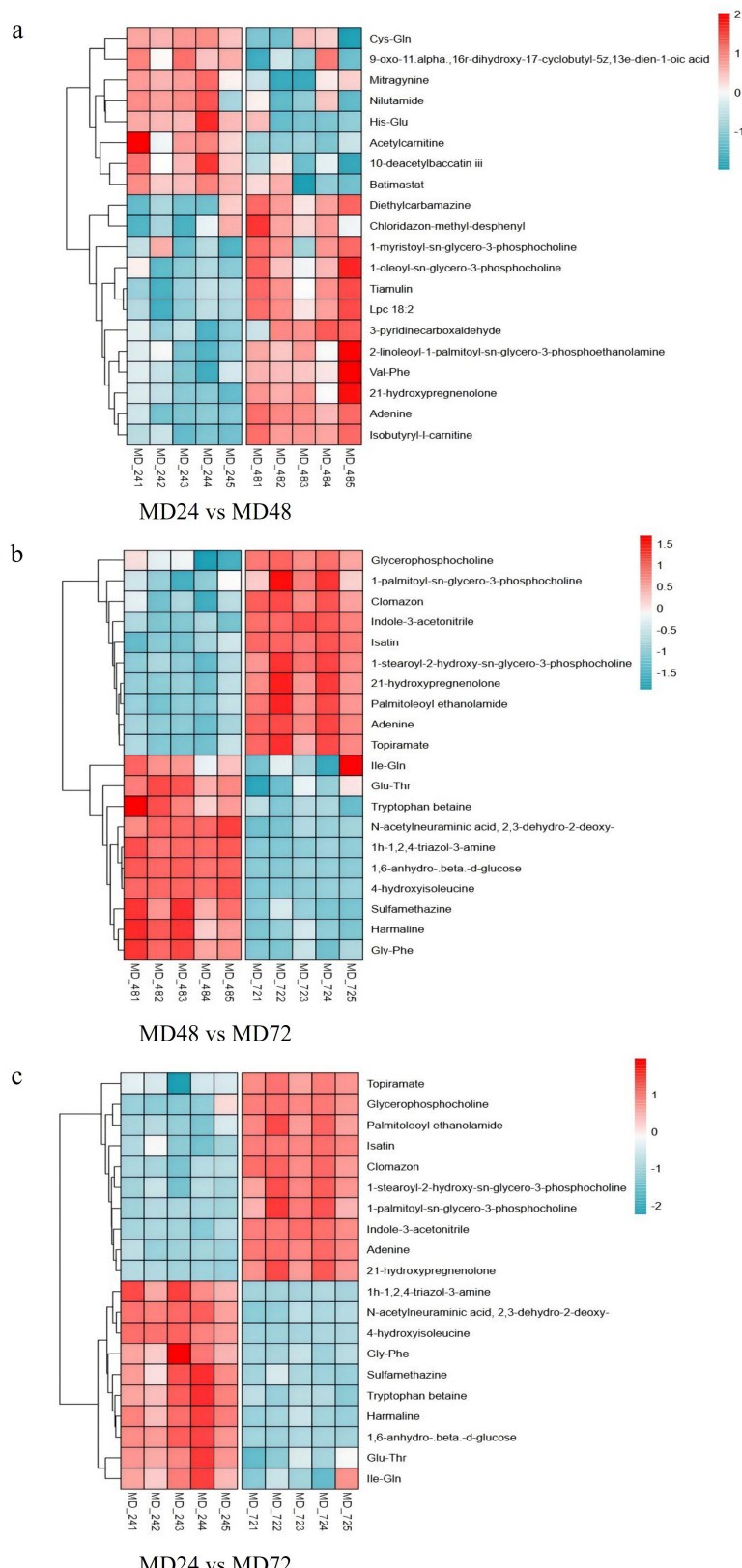

**FIG 6** The clustering tree diagram represents the top 20 discriminating metabolites in each test group, that is, MD24 vs MD48 (a), MD48 vs MD72 (b), and MD24 vs MD72 (c). Every row denotes a metabolite, and every column a sample. Color intensity reflects the concentration of each metabolite (red for

**Fig 6 (Continued)**

upregulated and green for downregulated). The co-culture metabolite data set was filtered using VIP > 1 and $P < 0.05$.

## DISCUSSION

The BaP degradation rates achieved by *Bacillus haynesii* TM-41 (36.79%) and *Kluyveromyces marxianus* TD-3 (38.63%) in our study were comparatively lower than those reported for environmental isolates of Bacillus and yeast species (21, 22). Although the two strains in this study, isolated from a food source (kefir), exhibited inherently low tolerance to PAHs, the co-culture system proved to be a highly effective strategy for significantly enhancing their BaP degradation rate. Research by Bhattacharya et al. (16) demonstrated the superior performance of co-cultures over pure culture in degrading BaP. Compared to the pure culture of *P. ostreatus* PO-3 (64.3%), co-culturing this strain with *Penicillium chrysogenum* MTCC 787 and *Pseudomonas aeruginosa* MTCC 1688 resulted in substantially higher degradation rates of 86.1% and 75.1%, respectively. When evaluated at 50 mg/L BaP, the co-culture of *Penicillium* sp. and *S. marcescens* achieved a degradation rate (65%) that was significantly more than double that of the bacteria in pure cultures (22). Given that kefir itself is a microbial assemblage comprising various microorganisms, including yeast, *Lactobacillus* sp., and *Bacillus* sp., there was an evident correlation between its microorganisms and the co-culture system constructed from it that is more conducive to the growth of its microorganisms. Consequently, a co-culture system of TM-41 and TD-3 was established for the degradation of BaP, achieving a degradation rate of 44.54% at 78 h, which was significantly higher than that in pure culture.

Following the optimization of abiotic factors by RSM, the BaP degradation rate was increased to 71.08%, which was 26.54% higher than that before optimization. As a well-known parameter affecting microbial growth, pH can influence the fungal-bacterial relationship by promoting or inhibiting the proliferation of one partner. Rousk et al. (23) investigated the regulatory role of pH on the competitive interaction between fungi and bacteria in soil. They found that a low pH favors fungal growth, while a high pH favors bacterial growth, and that bacteria exert competitive inhibition on fungi under high pH conditions. The optimal pH for growth and degradation identified in this study (pH 6) aligns with the findings of Huang et al. (24), who also reported pH 6 to be the most favorable condition. PHE removal decreased when the pH was lowered to 5 or 7, and 9. Similarly, the growth and degradation in the present study were also weak when the pH was lower or higher than 6, which indicated that the highly acidic and alkaline conditions were not particularly suitable for the co-culture system. Most strains exhibit optimal degradation activity under neutral to weakly alkaline conditions (25). Besides, the rate of microbial activities increased with temperature and reached its maximum level at an optimum temperature (26). A total of 12 yeast strains, isolated from various sources (wastewater, activated sludge, crude oil, and crude oil-contaminated soil) of an oil refinery, demonstrated the ability to degrade naphthalene, phenanthrene, pyrene, and crude oil at 27°C (9). The *Bacillus subtilis* strain, newly isolated from oil-contaminated soil, demonstrated optimal activity at 37°C. Within the range of 30°C–42°C, its microbial activity showed a clear pattern of increasing to a peak at 37°C, followed by a subsequent decline (27). In our study, although the optimal growth temperatures for TD-3 and TM-41 differ, the results of response surface optimization indicate that both strains show strong growth at 32°C.

OPLS-DA of pure culture and co-culture metabolites showed that the metabolites of the co-culture system were similar to those of TD-3 (Fig. S2). Herein, we discussed further the variations in the metabolites of main metabolisms, such as amino acids and lipid metabolisms, under pure and co-culture systems. The variation of amino acid and carbohydrate metabolisms promotes the expression of PAH degradation genes (28,

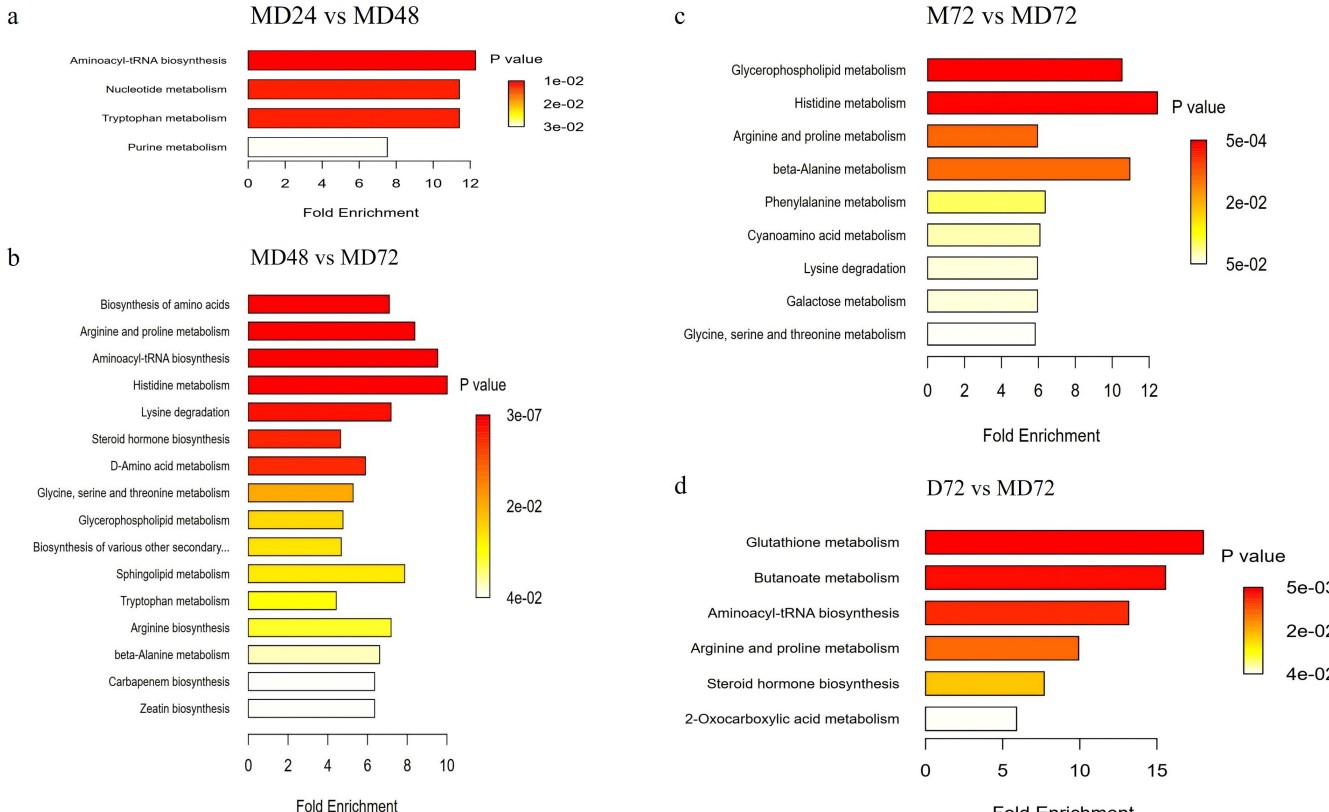

**FIG 7** Metabolite set enrichment analysis. (a) Differential metabolic enriched pathway in the co-culture at 24 h and 48 h; (b) differential metabolic enriched pathway in the co-culture at 48 h and 72 h; (c) differential metabolic enriched pathway in the TM-41 pure culture and co-culture at 72 h; and (d) differential metabolic enriched pathway in the TD-3 pure culture and co-culture at 72 h.

29). Concurrently, the biosynthesis of lipids (steroids) also interacts with the degradation of aromatic compounds (30). The metabolites of amino acid metabolism, that is, histidine metabolism, arginine and proline metabolism, and beta-alanine metabolism were significantly enhanced in the D72 vs MD72 group. In contrast, substantial changes in steroid hormone biosynthesis were noted in the D72 vs MD72 group. Their upregulation specified that material transformations in microbes may be enhanced under the co-culture system. Therefore, it can be inferred that during the co-culture microbial degradation of BaP, based on the metabolic complementarity between the two strains, TD-3 catabolizes BaP to provide energy for its partner, TM-41. Conversely, TM-41 synthesizes and secretes specific amino acids, which are assimilated by TD-3 to fulfill its nutritional demands for growth. Studies have shown that the degradation of PAHs is a sequential process carried out by fungi and bacteria, in which fungi initiate the attack on high-molecular-weight PAHs, and the resulting metabolic products are then subsequently mineralized by bacteria (31).

## Conclusions

This study showed that co-cultured microorganisms have great potential to enhance the degradation of BaP up to 71.08%. Moreover, the non-targeted metabolomic profile displayed a substantial variation in the metabolites of amino acids and lipid metabolism due to the synergistic effect of the co-cultured microorganisms. This study offers valuable insights into the synergistic mechanism of co-cultured microorganisms for the degradation of BaP and highlights the potential of food microbial strain resources for the removal of BaP.

## ACKNOWLEDGMENTS

This work was supported by the Natural Science Foundation of Xinjiang Uygur Autonomous Region (Grant number 2022D01D42) and the Natural Science Foundation of Xinjiang Uygur Autonomous Region (Grant number 2022D01C404, 2023B02034-1).

Q.W. and Z.Y. wrote and revised the whole manuscript. D.T. participated in the response surface optimization experiments. Q.L. participated in the liquid chromatography data analysis; B.Z. and J.X. directed the whole experiment and polished manuscripts. R.Z. and Y.Q. were responsible for writing, revising, and finalizing the manuscript. All authors read and approved the final manuscript.

## AUTHOR AFFILIATIONS

[1]Xinjiang Key Laboratory of Special Species Conservation and Regulatory Biology, College of Life Science, Xinjiang Normal University, Urumqi, Xinjiang, China
[2]Xinjiang Key Laboratory of Biological Resources and Genetic Engineering, College of Life Science & Technology, Urumqi, Xinjiang, China

## AUTHOR ORCIDs

Qing Wang ⓘ http://orcid.org/0009-0004-0009-1018
Rui Zhang ⓘ http://orcid.org/0009-0006-2630-6739
Yanan Qin ⓘ http://orcid.org/0000-0001-8240-6036

## FUNDING

| Funder | Grant(s) | Author(s) |
|---|---|---|
| Natural Science Foundation of Xinjiang Uygur Autonomous Region (Xinjiang Natural Science Foundation) | 2022D01D42 | Rui Zhang |

## AUTHOR CONTRIBUTIONS

Qing Wang, Conceptualization, Data curation, Methodology, Writing – original draft | Zhuonan Yang, Conceptualization, Data curation, Writing – original draft | Bei Zheng, Methodology, Supervision | Dilbar Tursun, Validation, Visualization | Qianjin Lv, Validation, Visualization | Jun Xin, Funding acquisition, Methodology, Project administration, Supervision, Writing – review and editing | Yanan Qin, Funding acquisition, Methodology, Project administration, Supervision, Writing – review and editing.

## DATA AVAILABILITY

The data reported in this paper have been deposited in the OMIX, China National Center for Bioinformation/Beijing Institute of Genomics, Chinese Academy of Sciences (https://ngdc.cncb.ac.cn/omix: accession no. OMIX011690). Strains TM-41 and TD-3 are deposited with access numbers PX225962 and PX210509 in GenBank.

## ADDITIONAL FILES

The following material is available online.

### Supplemental Material

**Supplemental figures and tables (Spectrum01840-25-s0001.docx).** Figures S1 and S2; Tables S1 and S2.

### Open Peer Review

**PEER REVIEW HISTORY (review-history.pdf).** An accounting of the reviewer comments and feedback.

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
