## [Reviewer comments · Microbiology Spectrum]

Microbiology Spectrum

Biodegradation of benzo(a)pyrene by division of labor in co-culture of *Bacillus haynesii* and *Kluyveromyces marxianus* from kefir

Qing Wang, Zhuonan Yang, Bei Zheng, Dilbar Tursun, Qianjing Lv, Jun Xin, Rui Zhang, and Yanan Qin

Corresponding Author(s): Rui Zhang, Xinjiang Normal University - Wenquan Campus

Review Timeline:

Submission Date:	June 12, 2025
Editorial Decision:	July 7, 2025
Revision Received:	September 15, 2025
Accepted:	September 24, 2025

Editor: Dan Li

Reviewer(s): Disclosure of reviewer identity is with reference to reviewer comments included in decision letter(s). The following individuals involved in review of your submission have agreed to reveal their identity: Doug Cossar (Reviewer #2)

Transaction Report:

DOI: <https://doi.org/10.1128/spectrum.01840-25>

Re: Spectrum01840-25 (**Biodegradation of benzo(a)pyrene by division of labor in co-culture of *Bacillus haynesii* and *Kluyveromyces marxianus* from kefir**)

Dear Prof. rui zhang:

Thank you for the privilege of reviewing your work. Below you will find my comments, instructions from the Spectrum editorial office, and the reviewer comments.

Revision Guidelines

Sincerely,
Dan Li
Editor
Microbiology Spectrum

Reviewer #1 (Comments for the Author):

Summary:

The authors compared the biodegradation rate of benzo(a)pyrene using mono- and co-culture of *Bacillus haynesii* and *Kluyveromyces marxianus* isolated from kefir and determined optimal culture conditions. Although the subject of the study is of interest, the manuscript is poorly written, and the result presentation and discussion were mostly confusing. In addition, the

study may not be relevant to food safety as presented in the manuscript.

Specific Comments:

Importance of study: Will processed foods be treated with yeast and bacteria to remove the benzo(a)pyrene formed during high temperature? Authors should discuss the feasibility and efficiency of using microbes to remove benzo(a)pyrene from food. Have the authors considered the relevance of the study findings for environmental purposes?

Line 67: What substrates are referred to?

Line 100-102: Authors should provide details of how benzo(a)pyrene was added to the minimal salt medium. Delete period after benzo(a)pyrene on line 102.

Lines 122-136: The solubility of benzo(a)pyrene in water is about 0.002 mg/L but 20 mg/L was the concentration used for cultures in this study. Authors should provide details on how benzo(a)pyrene was added to the cultures, if it was fully soluble in the culture, and how representative the samples collected for HPLC analysis are? For sample preparation, was the whole culture extracted or portions of it?

Lines 148- 149: Was the biomass measured using HPLC?

Lines 158-167: Authors should provide a clear description of the procedure used for metabolite extraction and detection by addressing the following questions-

- Why were only metabolites extracted from cells analyzed?
- What phase of monocultures was analyzed?
- Were samples collected at time intervals (24 h, 48 h, and 72 h) from both monocultures and co-cultures, from the re-suspended cells, or culture medium?
- Is the analysis described for UHPLC-QE-MS/MS or LC-MS analysis?

Line 180: Were there two strains of fungi and bacteria or one of each?

Lines 181-186 and Figure 1: Each photo should have a different label for easy reference.

Lines 197-205: How was the rate of biodegradation calculated? What phenomenon is referred to and why is it attributed to reduced OD for TD-3?

Lines 270 -284 and Figures 5 and 6: Define the treatment groups.

Figure 3 caption: "on degradation of BaP by the co-culture system." Define the co-culture system referred to. Provide more descriptive captions for all figures.

Manuscript edits: Identify and edit all typographical and grammatical errors in the manuscript. For example,

- Lines 43, 63, 147, 155, etc.: Correct all occurrence of "Bap" to "BaP".
- Line 57: Delete "decreasing"
- Lines 64-66: Re-word to clarify statement
- Line 68: Change "the co-culture" to "co-culture"
- Line 175: Revise sentence to clarify meaning.
- Line 239: Correct "The suggested the model"
- Lines 327-329: Correct spelling of metabolism
- Figure S1 caption: Change "BPA" to "BaP".

Reviewer #2 (Public repository details (Required)):

Metabolomics data should be made publicly available

Reviewer #2 (Comments for the Author):

The authors report on enhanced reduction of Benzo(a)pyrenes by a co-culture of a bacterium and a yeast isolated from a Kefir fermentation. There is a considerable body of data presented to provide the beginnings of a mechanistic description as to how the two microbes interact. The manuscript needs significant work to edit out many cases of misspelling, incorrect grammar, and awkward use of language. The discussion section is too long, containing much information better suited to the introduction, and can be shortened considerably. The metabolomics section is also not as concise as it could be and is difficult to follow, especially since the identity of the various samples is not defined in the text. Finally, the experimental methods section is lacking some important information - including the composition of the Minimal Salts Medium.

Response to Reviewer 1 Comments

We are deeply grateful for your thorough review and the exceptionally constructive feedback provided. Your suggestions have been invaluable in strengthening the academic rigor and overall quality of our work. We have undertaken a comprehensive revision of the manuscript, incorporating your feedback point by point. A detailed account of the revisions is provided in the following section:

Comments:

Importance of study: Will processed foods be treated with yeast and bacteria to remove the benzo(a)pyrene formed during high temperature? Authors should discuss the feasibility and efficiency of using microbes to remove benzo(a)pyrene from food. Have the authors considered the relevance of the study findings for environmental purposes?

Response: We sincerely appreciate the reviewer's attention to this issue. Benzo(a)pyrene is readily generated during high-temperature food processing, and microbial degradation is a promising method for its decontamination. Among the influencing factors, temperature is a critical one that affects the degradation efficiency. High temperatures can lead to the death or dormancy of microorganisms; in particular, the key enzymes responsible for degradation can denature and lose their activity, thereby completely eliminating their degradative capacity. In both current research and practical applications, the optimal temperature for microorganisms that can effectively degrade BaP typically ranges from 25°C to 45°C, with 30°C to 35°C being the temperature range at which many strains exhibit the highest activity.

A study by Zhang et al. found that two *Lactobacillus* strains are capable of adsorbing BaP from high-temperature processed meat products. Among them, *Lactobacillus plantarum* 121 achieved an adsorption rate of up to 41.21% for BaP in directly smoked sausages. However, it should be noted that the entire adsorption experiment was also conducted at 37°C. Bartkiene et al. studied the feasibility of degrading benzo[a]pyrene in cold smoked pork sausages using lactic acid bacteria (LAB). The results showed that: The treatment of sausages surface with LAB

(propagated in potato juice media) before smoking decreased the content of cadaverine and spermidine, whereas the treatment of sausages surface with LAB after smoking decreased the content of putrescine (approx. 53% when *L. sakei* and *P. acidilactici* were applied) or totally eliminated (applying *P. pentosaceus*) from outer layers and centre of sausages. The application of LAB for sausages treatment before and after smoking had significant influence on benzo[a]pyrene and chrysene decreasing.

Although this study primarily focuses on the application of kefir microorganisms in reducing foodborne benzo[a]pyrene (BaP), we recognize that its findings also hold significant relevance in the environmental field. Particularly in the food processing industry, by-products rich in polycyclic aromatic hydrocarbons (PAHs) are generated during the smoking and grilling of meat products. These substances may enter the environment through waste discharge or atmospheric release. Therefore, the application of kefir microorganisms as an intervention strategy can effectively reduce PAH emissions at the source, offering a potential biocontrol approach to mitigating environmental pollution from industrial activities.

We once again thank the reviewer for their valuable feedback and believe that these revisions broaden the research perspective and enhance its applicability, thereby increasing the overall impact of the paper.

ZHANG Zhuo, LI ZiQiang, QIAO YaFei, PEI JiaWei, ZHANG BoLin. Potential of *Lactobacillus* strains to bind benzo(a)pyrene in simulated meat products. *Microbiology China*, 2019, 46(9): 2345-2352. <https://link.cnki.net/doi/10.13344/j.microbiol.china.190094>

Comments: Line 67: What substrates are referred to?

Response: We sincerely appreciate the reviewer's attention to this issue. We apologize for the confusion caused by the previous inaccurate description. In the manuscript, the term "their substrates" specifically refers to benzo[a]pyrene. We have rephrased the original sentence for clarity: "Benzo[a]pyrene is not the first choice of carbon source for microorganisms, as it requires high energy to utilize benzo[a]pyrene, which limits the degradation of microorganisms to benzo[a]pyrene". Thank you again for your careful review, which helps improve the rigor of our work.

Comments: Line 100-102: Authors should provide details of how benzo(a)pyrene was added to the minimal salt medium. Delete period after benzo(a)pyrene on line 102.

Response: We sincerely appreciate your attention to detail and the suggestions provided to enhance the accuracy of our content. The question of how to add benzo[a]pyrene to the MSM medium was: First, a 1 g/L stock solution of benzo[a]pyrene in acetone was prepared. Subsequently, 2mL of the stock solution was added to 100 mL of Mineral Salt Medium (MSM) to obtain a final BaP concentration of 20 mg/L. Besides, we have deleted period as advised. Thank you again for your valuable input.

Comments: Lines 122-136: The solubility of benzo(a)pyrene in water is about 0.002 mg/L but 20 mg/L was the concentration used for cultures in this study. Authors should provide details on how benzo(a)pyrene was added to the cultures, if it was fully soluble in the culture, and how representative the samples collected for HPLC analysis are? For sample preparation, was the whole culture extracted or portions of it?

Response: We sincerely appreciate the reviewer's attention to this issue. We have added the specific procedures in the "Materials and Methods" section: "Since concentrations of BaP are substantially greater than its water solubility, a 1 g/L stock solution of BaP in acetone was prepared by adding BaP to sterilized acetone solution at the ratio of 1000 mL of acetone per gram of BaP stored at -20 °C until used"(now on page 6). Thus, BaP was completely dissolved in the MSM culture medium. Subsequently, at various time intervals, 1 mL aliquots of the well-mixed culture medium were collected from the incubator shaker to measure the degradation rate. Thank you again for your valuable input, which helps refine our work.

Comments: Lines 148- 149: Was the biomass measured using HPLC?

Response: We sincerely appreciate your valuable question regarding the "**biomass measure**" mentioned in the manuscript. We apologize for the confusion caused by the previous inaccurate description. The original sentence in the manuscript has been revised to read as follows:"The degradation of benzo[a]pyrene was measured by

HPLC with 1 mL of culture taken at different times (0, 8, 16, 24, 48, 72, 96 and 120 h) in pure culture and co-culture systems. The optical density at 600 nm of the strains were measured with a UV-visible spectrophotometer to determine cell growth, and OD600 was used to represent biomass" (now on page 8). Thank you again for your careful review, which helps improve the rigor of our work.

Comments: Lines 158-167: Authors should provide a clear description of the procedure used for metabolite extraction and detection by addressing the following questions:

- **Why were only metabolites extracted from cells analyzed?**
- **What phase of monocultures was analyzed?**
- **Were samples collected at time intervals (24 h, 48 h, and 72 h) from both monocultures and co-cultures, from the re-suspended cells, or culture medium?**
- **Is the analysis described for UHPLC-QE-MS/MS or LC-MS analysis?**

Response: We sincerely appreciate the reviewer's attention to this issue. We hereby provide a point-by-point response to the reviewers' valuable question.

Firstly, for the detection and identification of differential metabolites in the BaP degradation process, sampling was conducted at 24, 48, and 72 hours. At each time point, 10 mL of the liquid culture (**culture medium**) was collected from the following conditions: the pure culture of TM-41, the pure culture of TD-3, and the TM-41/TD-3 co-culture. All collected samples were immediately frozen and stored at -80°C for subsequent detection.

Secondly, the samples were slowly thawed at 4°C. A 100 µL sample from each liquid culture was extracted with 400 µL methanol/acetonitrile/water (2:2:1, v/v) solutions, followed by vortexing and a 30-minute low-temperature sonication. For protein precipitation, the samples were kept at -20 °C for 10 min and then centrifuged at 14,000g at 4 °C for 20 min. The supernatant was collected and vacuum-dried. Further, the dried extract was reconstituted in 100 µL of acetonitrile/water (1:1, v/v), vortexed, and centrifuged at 14,000g at 4 °C for 15 min to obtain the clear supernatants and then transferred to sample vials for further analysis.

Finally, the analysis was performed using UHPLC-QE-MS/MS and the detailed

chromatographic and mass spectrometric conditions were supplemented in the Materials and Methods section.(now on page 10)

We sincerely thank you for your valuable suggestions, which have provided important guidance for us to further improve research details, enhance the scientific rigor of our work, and play a crucial role in improving the quality of the study.

Comments: Line 180: Were there two strains of fungi and bacteria or one of each?

Response: We sincerely appreciate the reviewer's meticulous attention to this issue. The original sentence in the manuscript has been revised to read as follows: "A bacteria and a fungi strain were obtained from kefir that degrade benzo[a]pyrene were named TM-41 and TD-3, respectively". Thank you again for your valuable input.

Comments: Lines 181-186 and Figure 1: Each photo should have a different label for easy reference.

Response: We sincerely appreciate the reviewer's meticulous attention to this issue. We have completed the revision of this problem. (Please refer to Fig. 1)

Figure 1 Colony of the strain TM-41 (a) and TD-3 (b) on MRS medium; (c) Microscopic gram staining of strain TM-41; (d) Microscopic methylene blue staining of strain TD-3; (e) Neighbour-joining phylogenetic trees based on 16S rDNA sequences of strain TM-41; (f) The phylogenetic tree based on ITS sequences of strain TD-3.

0.01

0.01

Comments: Lines 197-205: How was the rate of biodegradation calculated? What phenomenon is referred to and why is it attributed to reduced OD for TD-3?

Response: We sincerely appreciate your valuable question regarding the

"phenomenon" mentioned in the manuscript. There was an error in the previous description, and we apologize for the confusion caused. We have corrected the relevant content in the manuscript: "The degradation of benzo[a]pyrene by TM-41 and TD-3 pure culture and the co-cultures were assessed at a concentration of 20 mg/L (Fig. 2a). At 96 hours, the pure cultures had the highest biodegradation rate for benzo[a]pyrene, at 36.79% and 38.63%, respectively. However, the observed degradation potentials of the co-cultures ranged from 19.87% to 44.54%, and the highest level of degradation was achieved at the 72 h. The co-culture significantly enhanced benzo[a]pyrene biodegradation in comparison with the pure TM-41 or TD-3 culture. As shown in Fig. 2b, strain TM-41 and strain TD-3 reached the maximum OD₆₀₀ values of 1.04 and 1.27 at 120 h and 96 h, respectively. Compared to pure culture, the growth of co-cultured microorganisms continuously increased during the reaction period and remained higher than that of pure culture. Thus, corroborating the highest benzo[a]pyrene biodegradation (44.54%) obtained for the co-culture system". And the formula for calculating the biodegradation rate was added to the 'Biodegradation studies' subsection of the Materials and Methods section. Thank you again for your careful review, which helps improve the rigor of our work.

Comments: Lines 270 -284 and Figures 5 and 6: Define the treatment groups.

Response: We sincerely appreciate your attention to detail and the suggestions provided to enhance the accuracy of our content. The details of the samples are defined in the "Materials and Methods" section of the manuscript: "5 mL of liquid culture from the pure and co-cultures of TM-41 and TD-3 were collected and stored at -80°C. Samples collected at the early (24 h), middle (48 h) and late (72 h) stages of degradation were designated as M24, D24, MD24; M48, D48, MD48; and M72, D72, MD72, respectively".

Additionally, the captions for Figure 5 and Figure 6 have been revised as follows: "**Figure 5** (a) The Partial least squares discriminant analysis (OPLS-DA) score plot show a clear separation degree of the three groups of samples. (b) Principal coordinate analysis (PCA) also demonstrates separation degree in all the group. (c-e) The horizontal coordinate of volcano graphs represents the value of fold change of

metabolite expression difference between two groups, i.e., \log_2FC , and the vertical coordinate is the value of statistical test of metabolite expression difference, i.e., $-\log_{10}(p_value)$ value, the higher the value the more significant the expression difference, the values of horizontal and vertical coordinates are logarithmized. The points on the green colored are metabolites with downregulated expression differences, and the points on the red colored are metabolites with upregulated expression differences" and "Figure 6 The clustering tree diagram represents the top 20 discriminating metabolites in each test group i.e., MD24 vs MD48 (a), MD48 vs MD72 (b) and MD24 vs MD72 (c). Every row denotes a metabolite, and every column a sample. Color intensity reflects the concentration of each metabolite (red for upregulated, green for downregulated). The co-culture metabolite data set was filtered using $VIP > 1$ and $P < 0.05$ ". Thank you again for your valuable input.

Comments: Figure 3 caption: "on degradation of BaP by the co-culture system." Define the co-culture system referred to. Provide more descriptive captions for all figures.

Response: We sincerely appreciate your attention to detail and the suggestions provided to enhance the accuracy of our content. We have completed the revision of this problem, including Figure 3 caption: Effect of initial BaP concentration(a),temperature(b), pH (c) and time(d) on degradation of BaP by the TD-3 and TM-41 co-culture system. Thank you again for your valuable input, which helps refine our work.

Response to Reviewer 2 Comments

Thank you very much for your comments and professional advice. These opinions help to improve academic rigor of our article. Based on your suggestion and request, we have made-corrected modifications on the revised manuscript. The specific revisions are as follows:

Comments 1: Metabolomics data should be made publicly available

Response: We sincerely appreciate the reviewer's attention to this issue. We have completed the revision of this problem. The data reported in this paper have been deposited in the OMIX, China National Center for Bioinformation / Beijing Institute of Genomics, Chinese Academy of Sciences (<https://ngdc.cncb.ac.cn/omix>: accession no. OMIX011690).

Comments 2: The manuscript needs significant work to edit out many cases of misspelling, incorrect grammar, and awkward use of language. The discussion section is too long, containing much information better suited to the introduction, and can be shortened considerably. The metabolomics section is also not as concise as it could be and is difficult to follow, especially since the identity of the various samples is not defined in the text.

Response: We are sincerely grateful for your meticulous review of our manuscript. We have corrected the spelling and grammatical errors throughout the manuscript and polished the language for clarity and flow. Additionally, we relocated the description of Discussion section: “[The genera *Candida*, *Cryptococcus*, *Pichia*, *Rhodospiridium*, *Rhodotorula* and *Saccharomyces* were considered potential degraders for harmful organic contaminants (9).]” and “[Sultana et al. (11) isolated five probiotic species, each with unique morphologies and the ability to tolerate benzo(a)pyrene, from a collection of 26 fermented foods. Among these, *Bacillus velezensis* PMC10 demonstrated the highest degradation efficiency for BaP, reaching 51.32%.]” to the Introduction section.

We have organized the discussion section (now on Page 19-21). Besides, we have provided a concise analysis of the metabolomics data in the Results section (now

on Page 11-18). We have also supplemented the Materials and Methods section with specific information for each sample: “5 mL of liquid culture from the pure and co-cultures of TM-41 and TD-3 were collected and stored at -80°C. Samples collected at the early (24 h), middle (48 h) and late (72 h) stages of degradation were designated as M24, D24, MD24; M48, D48, MD48; and M72, D72, MD72, respectively”. Thank you again for your careful review, which helps improve the rigor of our work.

Comments 3: The experimental methods section is lacking some important information - including the composition of the Minimal Salts Medium.

Response: We sincerely appreciate your insightful feedback on our experimental methods. Accordingly, we have reorganized the ‘Materials and Methods’ section, including the details of the chemicals and the composition of the minimal salt medium. For example: “Benzo[a]pyrene (>99% purity), acetone, dichloromethane, methanol (>99% purity). All chemicals were of analytical grade and purchased from Beijing Dingguo Changsheng Biotechnology Co., Ltd. The kefir grains used in this study were collected from a local household in the Kashgar, Xinjiang (37°46'24"N, 75°13'27"E). Minimal Salt Medium (MSM, pH 7.0) contained NH_4NO_3 1.00 g/L, $\text{MgSO}_4 \cdot 7\text{H}_2\text{O}$ 0.20 g/L, KH_2PO_4 0.50 g/L, K_2HPO_4 1.50 g/L, NaCl 0.50 g/L, $(\text{NH}_4)_2\text{SO}_4$ 0.50g/L”(now on Page 5). Thank you again for your valuable input.

Response to Manuscript edits Comments

We are sincerely grateful for your meticulous review of our manuscript. We have carefully revised the manuscript to fix all typographical and grammatical errors. The specific changes are as follows:

Comments- Lines 43, 63, 147, 155, etc.: Correct all occurrence of "Bap" to "BaP"

Response: This has been corrected in the manuscript. All occurrence of "Bap" in the manuscript have been changed to "BaP".

Comments- Line 57: Delete "decreasing"

Response: The original sentence has been revised to: "Bartkiene et al. found that the application of lactic acid bacteria for sausages treatment before and after smoking significantly decreased both benzo[a]pyrene and chrysene".

Comments- Lines 64-66: Re-word to clarify statement

Response: To improve clarity, this sentence has been amended as follows: "However, benzo[a]pyrene is not the first choice of carbon source for microorganisms, as it requires high energy to utilize benzo[a]pyrene, which limits the degradation of microorganisms to benzo[a]pyrene. Existing strains exhibit low degradation efficiency for benzo[a]pyrene and fail to achieve its complete mineralization".

Comments- Line 68: Change "the co-culture" to "co-culture"

Response: The original sentence has been revised to: "Compared with pure culture, co-culture has more advantages, especially the metabolic diversity...".

Comments- Line 175: Revise sentence to clarify meaning

Response: To improve clarity, this sentence has been amended as follows: "All statistical analyses were performed using IBM SPSS Statistics, version 19.0. Student's t-test was used for comparisons between two groups, and one-way analysis of variance (ANOVA) was used for comparisons among multiple groups. The standard deviation was expressed by the error bars of three repeated experiments. The metabolome data were analyzed on the free online platform of Wekemo Bioincloud

(<https://www.bioincloud.tech>). The data reported in this paper have been deposited in the OMIX, China National Center for Bioinformatics / Beijing Institute of Genomics, Chinese Academy of Sciences (<https://ngdc.cncb.ac.cn/omix>: accession no. OMIX011690)." (now on Page 11).

Comments- Line 239: Correct "The suggested the model"

Response: The original sentence has been revised to: "The high determination coefficient R^2 of 0.9309 suggests that the model effectively captured around 93% of the responses, indicating strong agreement between the predicted and experimental values. In general, this model for benzo[a]pyrene degradation is highly significant ($p < 0.0001$), suggesting that the quadratic polynomial model developed for benzo[a]pyrene degradation by co-cultures was reliable and workable in representing the actual relationship between the response and variables" (now on Page 14).

Comments- Lines 327-329: Correct spelling of metabolism

Response: This has been corrected in the manuscript (now on Page 17-18)

Comments- Figure S1 caption: Change "BPA" to "BaP".

Response: Figure S1 has been replaced and an appropriate caption has been provided.

Re: Spectrum01840-25R1 (**Biodegradation of benzo(a)pyrene by division of labor in co-culture of *Bacillus haynesii* and *Kluyveromyces marxianus* from kefir**)

Dear Prof. Rui Zhang:

Your manuscript has been accepted, and I am forwarding it to the ASM production staff for publication. Your paper will first be checked to make sure all elements meet the technical requirements. ASM staff will contact you if anything needs to be revised before copyediting and production can begin. Otherwise, you will be notified when your proofs are ready to be viewed.

Sincerely,
Dan Li
Editor
Microbiology Spectrum

Reviewer #2 (Comments for the Author):

The authors have provided a reasonable response to previous review comments and the manuscript is significantly improved. Some questions may remain over utility and the transition from laboratory to practice, from a defined set of controlled conditions to a less well defined environment.